# The Clinical Relevance of Epithelial-to-Mesenchymal Transition Hallmarks: A Cut-Off-Based Approach in Healthy and Cancerous Cell Lines

**DOI:** 10.3390/ijms26083617

**Published:** 2025-04-11

**Authors:** Maria Cristina Rapanotti, Elisa Cugini, Maria Giovanna Scioli, Tonia Cenci, Silvia Anzillotti, Martina Puzzuoli, Alessandro Terrinoni, Amedeo Ferlosio, Anastasia De Luca, Augusto Orlandi

**Affiliations:** 1Anatomic Pathology, Department of Integrated Care Processes, University of Rome Tor Vergata, Viale Oxford 81, 00133 Rome, Italyscioli@med.uniroma2.it (M.G.S.); tonia.cenci@gmail.com (T.C.); anzillottisilvia@gmail.com (S.A.); martina.puzzuoli@gmail.com (M.P.); ferlosio@med.uniroma2.it (A.F.); orlandi@uniroma2.it (A.O.); 2Department of Laboratory Medicine, Tor Vergata University Hospital, 00133 Rome, Italy; alessandro.terrinoni@uniroma2.it; 3Department of Experimental Medicine, University of Rome Tor Vergata, Via Montpellier 1, 00133 Rome, Italy; 4Department of Biology, University of Rome Tor Vergata, Via della Ricerca Scientifica 1, 00133 Rome, Italy

**Keywords:** epithelial-to-mesenchymal transition, healthy cells, mesenchymal cells, cancerous cells

## Abstract

The atypical activation of the epithelial-to-mesenchymal transition represents one of the main mechanisms driving cancer cell dissemination. It enables epithelial cancer cells to detach from the primary tumor mass and gain survival advantages in the bloodstream, significantly contributing to the spread of circulating tumor cells. Notably, epithelial-to-mesenchymal transition is not a binary process but rather leads to the formation of a wide range of cell subpopulations characterized by the simultaneous expression of both epithelial and mesenchymal markers. Therefore, analyzing the modulation of EMT hallmarks during the conversion from healthy cells to metastatic cancer cells, which acquire stem mesenchymal characteristics, is of particular interest. This study investigates the expression of a panel of epithelial-to-mesenchymal transition-related genes in healthy cells, primary and metastatic cancer cells, and in mesenchymal cell lines, derived from various tissues, including the lung, colon, pancreas, skin, and neuro-ectoderm, with the aim of identifying potential cut-off values for assessing cancer aggressiveness. Interestingly, we found that the expression levels of *CDH1*, which encodes the epithelial marker E-cadherin, *CDH5*, encoding vascular endothelial cadherin, and the epithelial-to-mesenchymal transition-transcription factor *ZEB1*, effectively distinguished primary from metastatic cancer cells. Additionally, our data suggest a tissue-specific signature in the modulation of epithelial-to-mesenchymal transition markers during cancer progression. Overall, our results underscore the importance of investigating epithelial-to-mesenchymal transition as a tissue-specific process to identify the most suitable markers acting as potential indicators of disease aggressiveness and therapeutic responsiveness.

## 1. Introduction

The epithelial-to-mesenchymal transition (EMT) represents a physiological process occurring during embryonic development (Type I EMT) and wound healing (Type II EMT), but it is aberrantly activated in pathological circumstances, such as fibrosis and the dissemination of cancer metastases (Type III EMT) [1].

Typically, epithelial cells are characterized by cell–cell interactions mediated by adhesion molecules such as E-cadherin and cytokeratins within tight junctions, adherens junctions, desmosomes, and gap junctions. Apical–basal polarity is also a key epithelial feature. The activation of EMT led to the loss of cell junctions and of the apical–basal polarity of epithelial cells inducing the acquisition of a mesenchymal phenotype. This transition confers increased aggressiveness, motility, and invasiveness to cancer cells [2,3]. The gain of this great metastatic potential allows cancer cells to detach from the primary tumor mass, enter the bloodstream, and eventually extravasate to colonize distant organs. There, the mesenchymal-to-epithelial transition (MET) occurs, facilitating the successful establishment of cancer metastases [4,5,6]. Two main events occur during EMT, namely (i) the downregulation of the epithelial adhesive protein E-cadherin and of junctional proteins such as occludins and claudins, which leads to the disruption of cell–cell adhesion and changes in cell morphology, and (ii) the expression of mesenchymal genes including those coding for N-cadherin, vimentin, and fibronectin, which in turn drive cell locomotion and invasion [4,7]. The modulation of epithelial and mesenchymal genes is orchestrated by the EMT transcription factors (EMT-TFs) TWIST1, ZEB1/2, SNAI1, and SNAI2 activated upon the engagement of different receptors (e.g., TKR, TGFβ family receptors) from several stimuli, including growth factors (i.e., EGF) and cytokines (i.e., TGFβ). The fulfillment of EMT triggers the dissemination of different types of cancers [2,8]. Several studies have highlighted the role of the EMT-TF SNAI1 as an effective inducer of EMT, whilst ZEB and TWIST seem to stabilize the acquisition of the mesenchymal phenotype by cancer cells [9]. In non-small-cell lung cancer (NSCLC) [10,11] and in breast cancer, SNAI1 has been correlated with cancer aggressiveness [12]. In breast cancer, SNAI1 levels were also associated with both incidence of cancer relapse and poor patient survival [12]. This role of SNAI1 as main TF-driving EMT has been further supported in breast cancer stem cells, where SNAI1 successfully regulated the transcription of EMT-related genes [13]. In particular, SNAI1 and SNAI2, along with ZEB1/2, bind the promoter of the *CDH1* gene, repressing its expression [14]. TWIST1 is a helix–loop–helix protein that plays a role in transcriptional regulation during cell differentiation. Increased expression of TWIST1 has been observed in various types of tumor cells, including prostate, gastric, and breast cancer. Moreover, TWIST1 can repress E-cadherin expression while promoting the upregulation of N-cadherin, inducing the so called “cadherins switch” [15,16]. TWIST1 seems to be implied mainly in the modulation of the expression of mesenchymal genes, such as *VIM*. However, the roles of the EMT-TFs in cancer progression remain controversial, even performing distinct, non-redundant functions that complement each other [14].

Fibronectin is a hallmark of mesenchymal cells, and it is considered the main marker of EMT progression. Once secreted in the extracellular environment, fibronectin builds a scaffold useful for cancer cell dissemination and invasion in the surrounding stroma. Fibronectin is also required during the early stages of the neoangiogenic process, facilitating the delivery of nutrients to the growing tumor mass [17]. In colorectal cancer (CRC), fibronectin levels are positively correlated to tumor growth and drug resistance [18]; in melanoma, it supports cancer cell proliferation and metastasis formation by hampering apoptosis and inducing EMT [19]. Intriguingly, in melanoma, fibronectin is regulated by the BRAFV600E signaling cascade promoting cancer cell aggressiveness, whereas this regulation does not occur in healthy melanocytes [20]. In non-small-cell lung cancer (NSCLC), the increase in fibronectin in the extracellular matrix strongly contributes to tumor growth [20]. Vimentin belongs to the intermediate filament family of proteins, and it is highly expressed in many epithelial tumors, such as breast cancer, prostate cancer, melanoma, and lung cancer. Its expression is a determinant for tumor growth and invasion; it is correlated with poor prognosis and could serve as a potential target for cancer therapy [21]. Interestingly, in melanoma, it has been demonstrated that protein kinase C-iota (PKC-ι)-induced EMT concurrently upregulated vimentin expression [22]. In MCF7 breast cancer cells, the increase in vimentin levels improved cell stiffness and motility, resulting in the induction of EMT following the downregulation of E-cadherin [23]. However, it has also been reported that vimentin prevents the development of CRC in a mouse model of colitis-induced CRC [24]. Thus, the role of vimentin in cancer progression is still unclear. The EMT-TFs, besides boosting the metastatic skill of cancer cells, confer a survival advantage to the cancer cells detached from the primary tumor mass, known as circulating tumor cells (CTCs), once they entered the bloodstream. Indeed, EMT-TFs promote CTCs’ acquisition of a stem-like phenotype and resistance to apoptotic stimuli [4,5,25,26,27,28,29,30]. In this context, another crucial factor sustaining cancer cell spread is cytoskeletal reorganization, in which the altered expression of cell adhesion molecules promotes the activation of matrix metalloproteinases (MMPs), particularly the gelatinases MMP-2 and MMP-9, which degrade the basement membrane [31,32]. The increased levels of MMP-2 and MMP-9 in breast cancer are associated with the development of metastasis, poor survival, and larger tumor size [31]. In CRC, the expression levels of MMP-2 and MMP-9 in healthy mucosa are strongly correlated with a worse 5-year survival for patients [33]. In pancreatic cancer, MMP-2, expressed by activated pancreatic stellate cells, is associated with cancer progression [34] in contrast to MMP-9, which seems to coordinate immune cell infiltration [35].

A key process promoting, supporting, and providing essential micronutrients to facilitate cancer cell spread is angiogenesis. The vascular endothelial cadherin (VE-cadherin) is a transmembrane adhesion molecule expressed by endothelial cells [36]. However, VE-cadherin is expressed in highly aggressive melanoma to produce vasculogenic structures and not in less invasive phenotypes of melanoma [37,38]. Indeed, Hendrix and colleagues detected the expression of VE-cadherin only in highly aggressive cutaneous (C8161 cell line) and uveal melanoma cells (C918 cell line). The abrogation of CDH5 expression led to the loss of these cells′ ability to form vasculogenic networks [37]. In CRC, the levels of VE-cadherin correlated with those of vascular endothelial growth factor receptor 2 (VEGFR2 or KDR), prompting endothelial differentiation, metastasis/recurrence, and poor prognosis [39]. Also in pancreatic cancer, VE-cadherin, following binding of the EMT-TF TWIST1 to its promoter, modulates the formation of vasculogenic nets [40]. Recent findings demonstrate that cancer cells, as well as CTCs, rather than in a complete mesenchymal state, acquire a hybrid epithelial/mesenchymal phenotype [3], forming distinct cell “subpopulations” or conferring to CTCs different plasticity, invasiveness, and metastatic potential [41,42]. Indeed, the EMT process does not act to transform cells from a purely epithelial to a fully mesenchymal phenotype. Accordingly, CTCs exist in different hybrid states, characterized by the coexistence of both epithelial and mesenchymal or stem cell features, following the activation of several transcription factors, signaling pathways, and modulation of the microenvironment. The purpose of this hybrid phenotype is to enable CTCs to evade immune surveillance, extravasate, survive in the bloodstream, and ultimately form metastases in distant organs [43,44].

However, the molecular signature of these cells in the different types of tumors is not clear, and the description of the molecular attributes of CTCs could clarify the mechanisms underlying malignant spread. For example, in advanced metastatic breast cancer patients, the number of mitotic CTCs increased, promoting their survival [45]; in patients affected by progressive tumors, the fraction of mesenchymal CTCs increased.

We identified the melanoma cell adhesion molecule (MCAM/MUC18/CD146) and the ATP binding cassette subfamily B member 5 (ABCB5) as melanoma-specific targets to isolate and purify a highly primitive and aggressive subset of circulating melanoma cells (CMCs) [46]. We demonstrated that the presence of these cells in the blood of melanoma patients is associated with disease progression [46]. Thus, we investigated the EMT-related molecular signature of these hybrid stem mesenchymal CMCs (CD45^−^CD146^+^ABCB5^+^) as potential biomarkers of melanoma progression and/or aggressiveness [47]. We compared the expression levels of a panel of EMT-related genes obtained in the enriched CD45^−^CD146^+^ABCB5^+^ CMCs at the disease onset with those collected during checkpoint follow-up either after six months of targeted or immune therapy. Even if performed in a restricted cohort of patients and considering the low RNA input deriving from these rare cells, we established a reliable quantitative RT-PCR (qRT-PCR) protocol, showing high sensitivity and specificity, to characterize CMC gene expression. Our results demonstrate that the EMT profile of the purified CMCs correlated with patients′ clinical outcome, suggesting its potential clinical value. In particular, during active disease, enriched CD45^−^CD146^+^ABCB5^+^ CMCs, besides downregulating the *CDH1* gene, upregulated the mesenchymal genes *VIM, CDH2* and *MCAM/CD146*, and the EMT-TF gene *TWIST1*, overall supporting EMT activation as the cause for metastasis and resistance to chemotherapy [47].

In light of this evidence, we propose to (i) identify, among the EMT genes expressed across different types of neoplasia, the most sensitive, specific, and reliable genes for cancer diagnosis and for the evaluation of the risk of recurrence in patients and (ii) establish a putative experimental threshold of the selected EMT hallmarks. To this purpose, the necessary and preliminary “conditio sine qua non” is the analysis of the mRNA levels of all the mentioned biomarkers in healthy models and in different stages of cancer progression. As models, we used commercially available healthy human cell lines and compared them to primary and metastatic tumor cells obtained from different tissues of origin. We included in our study cells derived from the lung, colon, pancreas, skin, and neuro-ectoderm, as well as mesenchymal cell lines. Recently, the “EMT-ome” has been described through the generation of a database containing all the omic data regarding EMT collected in pan-cancer [48]. Nevertheless, we believe that the simultaneous in vitro study of a panel of EMT-related mRNA, in a spectrum of distinct cancers compared to the healthy counterpart, could represent a new experimental approach, providing insights into cancer invasion and metastasis and representing a helpful clinical tool for cancer patient stratification.

## 2. Results

### 2.1. Expression of a Subset of EMT Markers Defines Cancer Progression

To identify the EMT profile possibly describing the different stages of cancer progression, at first, we analyzed the expression levels of a wide panel of EMT hallmarks in healthy cell lines and compared them to primary and metastatic tumor cells, as well as to mesenchymal cell lines, independently from their tissue of origin. The cellular models investigated encompassed lung-, neuro-ectoderm-, colon-, pancreas-, epithelium-, endothelium-, and bone marrow-derived cells. Specifically, we included five healthy human cell lines, eight cell lines derived from primary tumor sites, five metastatic cancer cell lines, and seven stromal/mesenchymal human cell lines (see as the Table in Material and Methods section). The gene panel assessed included the genes coding for the epithelial marker *CDH1*, the EMT transcription factors (EMT-TFs *SNAI1*, *SNAI2*, *TWIST1,* and *ZEB1*), the mesenchymal markers *VIM*, *hFN1*, *CDH2*, *CDH5,* and *MCAM*, and the metalloproteinases *MMP2* and *MMP9*. The data obtained are presented in Figure 1. Thus, we assigned a score to the mRNA levels of each gene detected upon normalization with the housekeeping gene *ACTB2*, together with a color scale ranging from 0.01 a.u. (pale yellow, •) to over 3000 a.u. (pink, ••••••••) (as described in the Materials and Methods Section 4.1). To establish the putative reference values for the expression of each gene, we considered the healthy cell lines as our reference point. Since the healthy cell lines included were primarily epithelial, the cut-off values were determined by considering these cells as expressing high levels of epithelial markers (e.g., *CDH1*) and low levels of mesenchymal markers (e.g., *hFN1*, *CDH2*, and *MMPs*). Then, we used mesenchymal cells as our positive control for the mRNA expression of mesenchymal genes, including *hFN1*, *TWIST1*, *SNAI1*, *MCAM*, and *MMP2*. Then, we considered the mesenchymal cells as our positive control for the mRNA levels of mesenchymal genes (hFN1, TWIST1, SNAI1, MCAM, and MMP2). We found that the expression of *TWIST1*, *SNAI1*, *ZEB1*, *hFN1*, and *MMP2* increased from healthy epithelial cells to primary and metastatic tumor cells, reaching the highest expression in mesenchymal cell models. Interestingly, we could also argue a consistent decrease in the levels of the epithelial *CDH1* mRNA. On the contrary, *MMP9* seemed completely nonspecific and not correlated to an increase in cancer cell aggressiveness, while other genes such as *VIM*, *CDH5*, and *MCAM* showed a trend toward an increased expression in the metastatic counterpart, albeit not statistically significant.

To better define and visualize the possible changes occurring in the EMT hallmarks during the different phases of cancer, we also presented the results as bar graphs of the average level of each marker obtained in healthy epithelial cells, primary and metastatic tumors, and in the mesenchymal cells (Figure 2). This representation clearly shows a strong and significant downregulation of the epithelial marker *CDH1* in the primary cancer cell lines analyzed. Even though not statistically significant, *CDH1* reduction is also still clearly visible in the metastatic tumor cells and in the mesenchymal cells. *MCAM* seems to be associated preferably to some tumors such as primary epidermoid carcinoma, colon cancer, primary and metastatic malignant melanoma, and metastatic lung cancer.

Interesting to note is the upregulation of the EMT-TFs *TWIST1* and *SNAI1* strongly induced in the mesenchymal cell lines. Not statistically significant but showing an increasing trend is also the level of the EMT-TF *ZEB1*. We also observed the same trend for the mRNA of the mesenchymal marker *hFN1*, suggesting an increased aggressiveness of cells accompanied by an increase in *MMP2* but not *MMP9* mRNAs and of the *CDH5* gene, fundamental to sustain cancer cell proliferation through vasculogenic mimicry. Among them, *VIM* remained at high levels regardless of the cell model analyzed.

On the contrary, some genes, i.e., *CDH2* and *MMP9*, did not seem to respond to the progressive transformation of healthy cells into mesenchymal cells.

### 2.2. CDH1, ZEB1, and CDH5 Levels Could Discriminate Between Healthy and Cancer-Derived Cells

To investigate the possible accuracy of the EMT hallmark genes to discriminate between healthy and cancer patients, we performed a receiver operating characteristic (ROC) analysis. Although it is known that an accurate ROC analysis requires a much larger number of samples [49], we believe that this can still be a useful preliminary approach to identify possible diagnostic cut-offs to further deepen the study. As reported in Figure 3 and in Table 1, we found that in the samples analyzed, the *CDH1* mRNA levels could efficiently discriminate between healthy and primary tumor samples with 100% sensitivity and 90% specificity, with a cut-off range higher that 3.085 a.u. Also, the *CDH1/ZEB1* and ratio significantly differentiated healthy cells from primary tumor-derived cells with good sensitivity and specificity (Table 1).

The *ZEB1* gene allows for the distinction of healthy cells and cells obtained from metastatic tumors with 100% specificity and sensitivity and a cut-off higher than 0.2765 a.u. Although we did not observe any significant modulation in the mRNA levels of *CDH5* from healthy cells to mesenchymal ones (Figure 1 and Figure 2), *CDH5* mRNA levels could discriminate healthy and primary cancer-derived cells with 100% sensitivity and 80% specificity (cut-off > 0.135 a.u.); the *CDH5/CDH1* ratio differentiated both healthy from primary and metastatic cells with the respective cut-off > 0.001807 and >0.06031, whilst the *CDH5/ZEB1* ratio < 0.05273 also distinguished them (Table 1).

### 2.3. EMT Profile During Cancer Progression Is Tissue-Specific

To investigate the possible tissue-specificity of the EMT markers under investigation, the results obtained from the healthy, primary, and metastatic cancer cells models were divided based on the tissues of origin (Figure 4). As for Figure 4, we assigned a score and a color scale to ranges of mRNA levels.

In different tissues, we observed a distinctive EMT gene profile occurring during the neoplastic transformation. For instance, colon-derived cells showed a progressive increase in the levels of *SNAI2*, *ZEB1*, and *CDH5* starting from the healthy epithelial to the metastatic subtypes. In contrast, in neuro-ectoderm-derived cells, we observed an upregulation of the *TWIST1*, *ZEB1*, *CDH2*, *hFN1*, *CDH5*, and *MMP9* mRNA.

However, to better visualize these distributions, we also presented the collected data as bar graphs (Figure 5). The lack of statistical analyses is due to the number of cell lines analyzed in each group (healthy, primary tumor-derived, metastatic tumor-derived, and mesenchymal cells). Not all groups included at least three different cell lines, which is the minimum required for a reliable statistical analysis. However, through this representation, with all its limitations, it is possible to appreciate the different distribution of various EMT markers based on the tissue of origin of the cells used. In particular, we found that in the lung (Figure 5A) a trend toward an increase in *VIM*, as well as of the *CDH5*, *MCAM*, and *MMP2* mRNAs associated with the metastasis-derived cancer cell lines, whereas the *MMP9* gene seemed to be mainly related to the mesenchymal phenotype of these cells. Particularly, *MMP9* gene expression was found to be unremarkable both in primary and metastatic cells.

Interestingly, in the skin (Figure 5B) we found a shift toward a gradual decrease in *CDH1*, paralleled by a possible increase in the EMT-TFs *TWIST1* and *SNAI1* and of the *CDH5* gene. In this specific tumor subset *MMP9*, together with *MMP2* and *ZEB1*, were more related to the metastatic subtype of this tissue. In the colon-derived cells (Figure 5C), during the transition from a healthy to a tumor cell and then to a metastatic cell type, we observed a gradual increase in the EMT-TF genes *SNAI2* and *ZEB1*, and in *VIM*, *CDH5* and *MMP2*. Also, in neuro-ectodermic derived cells (Figure 5D), we found an upregulation of the EMT-TFs, but this time, it involves *TWIST1* and *ZEB1*, paralleled by an increase in the *hFN1* and *MCAM* mRNA. Of note, we detected a strong downregulation of the *CDH1* gene in the primary melanoma cells, whilst it came back in the metastatic cells. As for skin, *MMP2* appeared to be related to the metastatic cell subtype. In the pancreas-derived cells (Figure 5E), we found an upregulation of the EMT TFs *SNAI1, SNAI2*, and *ZEB1*. Nonetheless, it should be highlighted that the behavior of *SNAI2* seems to disappear on the primary tumor cells and be re-expressed in the metastatic and stromal-derived cells.

## 3. Discussion

The key role of EMT in cancer metastatization is determined by its implication in two fundamental steps of cancer cell dissemination, namely (i) the induction of cancer cell dissociation from the primary tumor mass following the acquisition of a mesenchymal phenotype and (ii) the gain of survival advantages to circulating tumor cells (CTCs), allowing them to successfully colonize the secondary site [50,51]. The implication of EMT in CTC generation and survival implies the production of a heterogeneous population of circulating cancer cells which frequently lacks epithelial markers, conversely expressing mesenchymal ones [3,52]. In breast cancer, mesenchymal CTCs have been associated with disease progression [53,54]. More generally, EMT phenotypes are commonly related to resistance to chemotherapy and immunotherapy, whilst the-mesenchymal-to-epithelial transition (MET) plays a key role in the final establishment of CTCs at the distant site, generating an epithelial metastasis [26,54].

Despite all these considerations, most of the currently available CTC isolation approaches are based on the epithelial nature of CTCs using the epithelial marker EpCAM to capture and enrich the CTC fraction [55,56].

We have previously established a reliable, highly sensitive, and specific quantitative real-time polymerase chain reaction (qRT-PCR) protocol to characterize the EMT gene profile of CD45^−^CD146^+^ABCB5^+^ circulating melanoma cells (CMCs), demonstrating the application of EMT hallmarks and EMT-TFs as potential biomarkers associated with primary tumor aggressiveness or response to therapy [46,47,57]. Indeed, notwithstanding the small cohort of patients collected, we found that the CMC expression levels of the genes coding for vimentin, TWIST1, and ZEB1 specifically differentiate patients responding to therapy from non-responder patients (doi: 10.3390/ijms241411792). These findings further support the potential clinical value of the analysis of the EMT gene signature in CTCs across different types of neoplasia, as well as melanoma. Thus, the next step involves defining the putative experimental threshold of selected EMT hallmarks (EMT-TFs *SNAI1*, *SNAI2*, *TWIST1,* and *ZEB1*), mesenchymal markers (*VIM*, *hFN1*, *CDH2*, *CDH5*, and *MCAM*), and the metallo-proteinases (*MMP2* and *MMP9*) that can be applied for cancer patients′ stratification based on cancer aggressiveness or response to therapies. To this extent, we compare the RNA levels of the above cited panel of genes obtained from commercially available healthy human cell lines and primary, metastatic, and mesenchymal cells from different tissues. We included cells from the lung, colon, pancreas, skin, and neuro-ectoderm. We found that genes involved in EMT—*CDH1*, *CDH5*, and *ZEB1*—allowed for the differentiation of healthy cells from tumor samples. Additionally, some other genes, such as *TWIST1*, *SNAI1*, *hFN1*, and *MMP2* reached the highest expression level in mesenchymal cell models. The ROC analyses, even performed on a limited number of samples, identified the cut-off levels of *CDH1*, *CDH5*, and *ZEB1* genes useful for the possible *diagnostic* identification of primary or metastatic tumors. The cut-off values for *CDH1* able to differentiate non-cancer from primary tumor cells is 0.385 a.u, whilst the *CDH1/ZEB1* and the *CDH5/CDH1* ratios discriminating healthy from primary tumor cells were <4.552 and >0.001807 a.u., respectively. *ZEB1* belongs to the EMT-TFs, and in colorectal cancer (CRC), a strong correlation between the cancer expression of ZEB1 and the number of CTCs detected in the blood of patients has been demonstrated [58]. This finding has been further supported in a paper showing that ZEB1 induction of Notch1 promotes the release of CTCs from lung squamous cell carcinoma, enhancing cancer aggressiveness [59]. Thus, our results add further details regarding the role of *ZEB1* in cancer diagnostics, suggesting its possible function in discriminating patients suffering from metastatic tumors from primary tumors. Additionally, *ZEB1* levels efficiently identified metastatic cancer cells for values higher than 0.2675 a.u. Also in this case, the *CDH5/CDH1* and *CDH1/ZEB* ratios differentiate healthy from metastatic cancer cells for a.u. values higher than 0.06031 and >4.186, respectively. The evidence of *CDH1* modulation in primary tumors is aligned with recent findings describing EMT as a multifaceted and often reversible process ranging from complete cell conversion toward a mesenchymal phenotype to an incomplete suppression of pre-existing epithelial characteristics and acquisition of mesenchymal features, originating from a variety of hybrid EMT intermediates [41,60]. Indeed, it has been reported that luminal breast cancer is mainly characterized by hybrid epithelial–mesenchymal cells contributing to its aggressiveness and that in basal breast cancer, the presence of hybrid EMT cancer cells is fundamental to conferring tumorigenicity to cancer cells [60,61,62]. In CRC, the hybrid EMT phenotype activates fibroblast-inducing collagen remodeling [63] and in lung cancer, the presence of cancer hybrid EMT cells increases the chance of success of the metastatic cascade [64,65]. Similarly, CTCs exist in a series of hybrid states, maintaining both epithelial and mesenchymal or stem cell properties. The acquisition of these mixed phenotypes allows CTCs to survive, evade immune surveillance, and eventually colonize distant organs [3,41,44,66].

Besides *CDH1* and *ZEB1*, *CDH5* also showed promising results. In particular, *CDH5* levels higher than 0.135 and 0.1865 significantly distinguished between healthy and cancer cells and primary vs. metastatic cancer cells, respectively. *CDH5* is the gene coding for the vascular endothelial growth factor (VEGF) protein. The importance of VEGF in supporting the metastatic process in widely accepted; it promotes angiogenesis and vascular permeability and activates different signaling cascades promoting tumor development [67]. In CRC, VEGF expression correlated with lymph node metastases [68] and an increased expression of VEGF was found in metastatic lesions from ovarian cancer compared to primary tumor tissues [69]. Thus, our results further support the clinical significance of *CDH5*.

The data about *hFN1* expression are interesting. In our previous study [47], we did not document the *hFN1* gene expression in any of the investigated melanoma CTC samples, although it is well known to be upregulated during EMT and to be implicated in resistance to radio therapy in cancer. This result contrasts with what we have found here, namely that in neuro-ectoderm-derived cells, *hFN1* reached the maximus levels in metastatic melanoma cells. It should be hypothesized that the fibronectin expression is functional for the successful establishment of melanoma metastases rather than contributing to melanoma cells spreading in the bloodstream.

However, these findings, although promising, deserve further validation, starting with repeating these analyses on cancer patient-derived tissues.

Subsequently, we analyzed the EMT gene signature in each specific tissue included in this study. The analysis of the same data, considering both the overall expression profile of EMT markers across all included cell lines and the tissue-specific EMT signature, has once again highlighted the complexity of this phenomenon, in agreement with the existing literature on EMT, particularly its heterogeneity. Indeed, while the comprehensive analysis of EMT markers in our cellular models suggested the modulation of specific genes (i.e., *CDH1*, *SNAI1*, *TWIST1*, *hFN1*, and *MMP2*), the assessment of their tissue-specific expression revealed a distinct expression profile. We found that *MMP2* and *MMP9* were markedly increased during the transformation from healthy to mesenchymal cells in all the tissues analyzed, e.g., the lung, skin, colon, and neuro-ectoderm, except for the pancreas. In contrast, the EMT-TFs, as expected, were widely involved in this neoplastic transformation in all tissues with the exclusion of the lung. *SNAI1* gradually increased in skin and the pancreas, whilst *SNAI2* was mainly amplified in the pancreas and neuro-ectoderm. We found an upregulation of *TWIST1* in skin and the neuro-ectoderm, whilst *ZEB1* seemed to contribute to the neoplastic transition of cells derived from skin, the colon, and pancreas.

The upregulation of TWIST1 expression has been found in skin cancer patients, such as melanoma squamous cell carcinoma, sharing a positive family history of recurrence. Differential BMI1, TWIST1, and SNAI2 mRNA expression patterns correlate with malignancy type in a spectrum of common cutaneous malignancies, including basal cell carcinoma, squamous cell carcinoma, and melanoma [70]. *TWIST1* and *SNAI2* as mesenchymal markers have been implicated in aggressive behavior cancers [71,72]. *TWIST1* seems to be involved in keratinocyte proliferation, tumor promotion, and EMT in a *dose-dependent* manner [73,74].

Regarding the EMT hallmarks, we observe the same heterogenicity across the tissues considered. Whilst *CDH5* was modulated in all the tissues apart from the neuro-ectoderm and pancreas, *CDH1* was modulated only in the neuro-ectoderm. *VIM* showed a strong increase in the colon and lung. In neuro-ectoderm-derived cells, we confirmed the role of *MCAM* in increasing cell aggressiveness and invasive potential. In particular, *MCAM* seems to be associated preferably with some tumors such as primary epidermoid carcinoma, colon cancer, primary and metastatic malignant melanoma, and metastatic lung cancer [75]. Indeed, the role of MCAM in melanoma spread is well established [76], and we have already shown its potential clinical application for the capture and enrichment of a circulating melanoma cell (CMC) subpopulation characterized by an impressive invasive capacity [46,77,78,79]. Overall, we documented an increase in the expression levels of selected EMT genes, from healthy epithelial cells to primary and metastatic tumor cells, specific to each tissue under investigation.

The results obtained in this study aimed to identify putative cut-off values for EMT-related genes that could be used to evaluate cancer aggressiveness and to better depict the tissues′ specific EMT. In addition to identifying some putative cut-offs, our data further confirmed that EMT cannot be considered in universal terms. Each specific tissue exhibits a unique signature, characterized by a hybrid E/M phenotype with more epithelial or mesenchymal features. Evidence has shown that cancer populations with partial EMT express the highest metastatic potential, particularly those sharing more epithelial features than mesenchymal [60]. Patients affected by primary prostate, breast, or lung cancers exhibiting a significant EMT plasticity have worse outcomes [4,8,26,30,80,81,82]. Our results further support this evidence, highlighting the need to study EMT as a tissue-specific process to identify the most appropriate markers that could serve as potential biomarkers for disease aggressiveness and possible therapeutic response.

On the contrary, other genes, such as *VIM* and *MMP9*, seemed completely nonspecific and not correlated to an increase in cancer cell aggressiveness.

## 4. Materials and Methods

### 4.1. Cell Lines

Cell lines, classified on the basis of their tissue of origin (lung, colon, pancreas, skin, neuro-ectoderm, endothelium, and bone marrow) and grouped depending on their subtype (healthy epithelial cells, primary tumor-derived cells, metastatic tumor-derived cells) were purchased from the American Type Culture Collection (ATCC, Manassas, VA, USA) (Table 2). Cells were grown in RPMI-1640 (GIBCO-Thermo Fisher Scientific, Waltham, MA, USA), with the exception of human stromal pancreatic cells (StromPa), and supplemented with 10% heat-inactivated fetal bovine serum (FBS), 2 mM glutamine, 100 U/mL penicillin, and 100 μg/mL streptomycin (GIBCO-Thermo Fisher Scientific, Waltham, MA, USA) in a humidified atmosphere at 37 °C, 5% CO_2_. StromPa were grown in PriGROWIII medium GIBCO-Thermo Fisher Scientific, Waltham, MA, USA). The cell lines included in the study were cultured independently three times. Cells were placed in cultures and subjected to a maximum of three passages and when cells reached about 70% of confluence, they were detached from the flasks following incubation with a solution of 0.5% trypsin–EDTA (EuroClone), washed twice with phosphate-buffered saline (PBS), and stored at –70 °C until RNA extraction. The different cell lines, purchased from ATCC, were *“cell-line”-authenticated* and certified for bacteria and virus absence. They were all subjected to a mycoplasma test (qPCR Kit Mycoplasma Testing, Carlo Erba-Dasit Group, Cornaredo, MI, Italy).

Cells were detached by trypsinization, centrifuged, washed twice with phosphate-buffered saline (PBS), and stored at −70 °C until RNA extraction.

Two distinct mesenchymal cell cultures (MSCs), originating from two bone marrow healthy donors, were included as fully mesenchymal cell models. To this aim, we used in-house isolated, expanded, and characterized MSCs from human bone marrow, obtained by applying a specific protocol for cell culturing. Immunofluorescence staining and Western blotting were used to specifically characterize the expression of differentiation markers and selected known markers [83].

We analyzed the following biomarkers: the epithelial marker *CDH1*, the EMT transcription factors (EMT-TF) *SNAI1*, *SNAI2*, *TWIST1*, and *ZEB1*, the mesenchymal markers *VIM*, *hFN1*, *CDH2*, *CDH5*, and *MCAM*, and the matrix metalloproteinases *MMP2* and *MMP9*, in benign and different stages of cancer progression of distinct neoplasms of our interest. The purpose of this study was the comparison of the expression levels of the mentioned hallmarks between distinct healthy cell lines to primary and metastatic tumor cells, as well as to mesenchymal cell lines.

### 4.2. Selection of Reference Genes Panel

The EMT candidate reference genes included in the panel were selected according to the literature (Pubmed databases). The selected hallmarks are known to be strongly associated with angiogenic potential, cell–cell adhesion molecule pathways, and matrix metalloproteinase extravasation; all processes are major key players in the regulation of EMT, early cancer spread, and disease progression. We included the key players of EMT, namely the genes coding for E-cadherin (*CDH1*), N-cadherin (*CDH2*), VE-cadherin (*CDH5*) involved in the “cadherin switch” during EMT [10,84,85], and the mesenchymal markers *VIM* (vimentin) and *hFN1* (fibronectin), defined as the core of EMT [7,10,42]. We added the EMT-TFs *TWIST1, SNAI1, SNAI2/SLUG*, and *ZEB1*, which orchestrate the gene expression rewiring occurring during EMT [86]. The endothelial antigen *MCAM/CD146* as a potential marker of cancer progression and metastasis driver [77,87] and the MMP2 and MMP9 matrix metalloproteinases were also included [31,32].

### 4.3. Quantitative Real-Time PCR Assay for Molecular EMT Profiling

Our main goal was to validate the previously established qRT-PCR protocol for the quantitative determination of the above reported EMT hallmarks [46] as a sensitive, specific, and reliable method for low-input diluted RNA to mimic a usual rare recovery from CTCs. To this aim, we tested no more than 100 ng of total RNA obtained from both epithelial, neuro-ectodermic, and mesenchymal healthy cell lines and their malignant counterparts.

Total RNA was extracted from all cell lines with TRIzol Reagent (Invitrogen, Waltham, MA, USA) according to the manufacturer′s instructions, using a home-made protocol based on Chomczyńsky and Sacchi’s method adapted for RNA extraction with an extremely low number of cells, including selective precipitation supported by glycogen [88]. RNA concentration and integrity was measured with ultraviolet spectrophotometry using the NanoDrop 2000 (Thermo Fisher Scientific, Waltham, MA, USA) according to the manufacturer′s instructions.

One hundred nanograms of RNA were reverse transcribed using Moloney murine leukemia virus (MMLV) reverse transcriptase (Promega, Madison, WI, USA). Afterwards, RT-qPCR was carried out based on SYBR Green chemistry (iTaq™ Universal SYBR^®^ Green Supermix, BIO-RAD, Hercules, CA, USA).

We assessed the analytical specificity and sensitivity of the RT-qPCR assay for the detection of the previous reported gene panel through optimization experiments in the included cell lines (Table 2). To this aim, we produced cDNA by reverse transcription (RT) of 100 ng of total RNA and then performed 1:5 serial dilutions of the resulting cDNA, starting from 10 ng to obtain calibration curves at 7 points. Specifically, we used 10, 2, 0.4, 0.08, 0.0032, and 0 ng of cDNA, in duplicate, to perform RT-qPCR assays for each investigated gene. Once we defined the calibration curves for each hallmark, we settled the best threshold (Ct), slope, and R2 (>0.96) value for each gene to be used for further experiments. Moreover, at the end of each RT-qPCR run, we performed melting curve optimization (60–95 °C) to verify the specificity of the reactions [46].

The RT-qPCR assay for the definition of the EMT profile of isolated CTCs was carried out in 20 μL of final volume containing 5 μL of retro-transcripted cDNA, 5 μM of each primer (Table 3), and 50% SYBR green (Kapa SYBR Fast qPCR kit; Kapa Biosystems, Roche, Wilmington, MA, USA) in a StepOnePlus Real-Time PCR System (Thermo-Fisher). The gene coding for β-actin, *ACTB*, was used as an internal reference gene. Duplicate reactions were run for each gene. For each sample, an amplification plot and corresponding dissociation curves were examined. Gene expression normalized to β-actin was calculated using the 2^−ΔCt^ method. The primer sequences used in the RT-qPCR analysis are listed in Table 3.

### 4.4. Statistical Analysis

Gene expression data were represented as violin plots. Statistical significance between healthy, primary and metastatic cancer-derived cells, and mesenchymal cells for each gene was evaluated by a non-parametric one-way ANOVA using the Kruskal–Wallis test for multiple comparisons. A *p*-value < 0.05 was considered significant. The diagnostic accuracy of the selected EMT mRNAs was assessed using the area under the curve (AUC) of receiver operator characteristics (ROC) curves, assessing their ability to distinguish healthy, primary, and metastatic cancer cells. The software used for statistical analysis was GraphPad Prism 10.1.1 (GraphPad Software, San Diego, CA, USA); a *p*-value < 0.05 was considered significant (* *p* < 0.05; ** *p* < 0.01; *** *p* < 0.001; **** *p* < 0.0001).

## Figures and Tables

**Figure 1 ijms-26-03617-f001:**
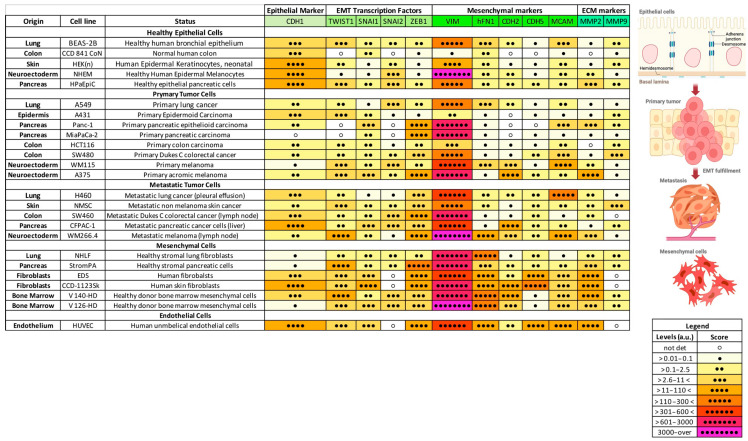
Correlogram of the relative expression levels of investigated EMT hallmarks obtained in healthy epithelial cells, primary and metastatic tumor cell lines, and in mesenchymal cells. The cell lines included in the study have been divided based on their tissue of origin (lung, colon, skin, neuro-ectoderm, endoderm, pancreas, endothelium, and bone marrow) and grouped depending on their subtype (healthy epithelial cells, primary tumor-derived cells, metastatic tumor-derived cells, and mesenchymal cells). The legend shows the magnitude of the expression levels assigned depending on the mRNA expression range obtained upon normalization of each sample with the ACTB2 gene together with the color scale. The drawings accompanying the table depict the transformation from healthy tissue to metastatic tumors, culminating in a fully mesenchymal phenotype. Created in BioRender.com.

**Figure 2 ijms-26-03617-f002:**
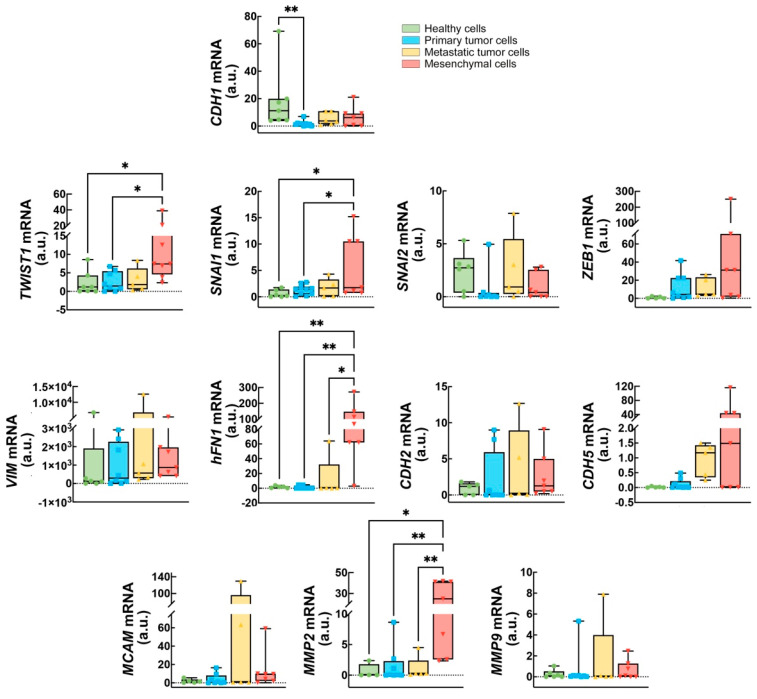
Averaged expression levels of each EMT hallmark across healthy epithelial cells, primary and metastatic cancer cell lines, and mesenchymal cells. The expression level of each gene is represented as the mean of the levels obtained in the cell lines analyzed. n > 4, one-way ANOVA, * *p* < 0.05, ** *p* < 0.005.

**Figure 3 ijms-26-03617-f003:**
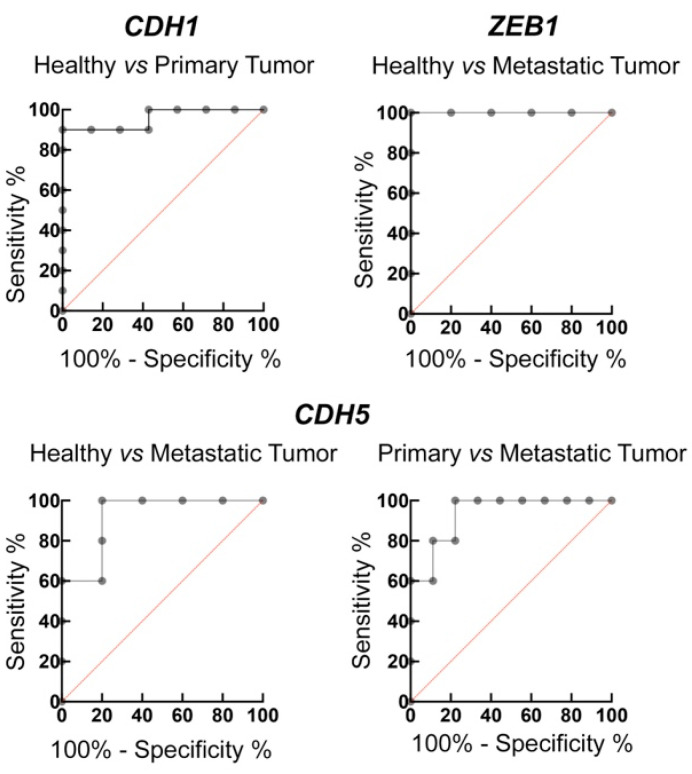
ROC curves of *CDH1*, *ZEB1*, and *CDH5* levels to discriminate cancer cells vs. healthy cells.

**Figure 4 ijms-26-03617-f004:**
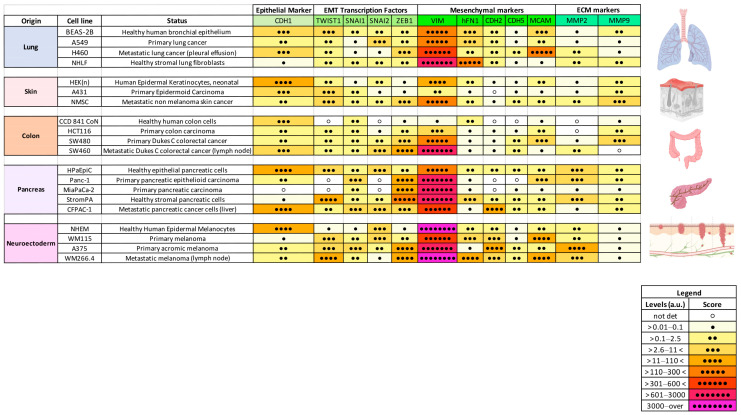
Correlogram of relative expression levels of investigated EMT hallmarks in healthy epithelial cells, cancerous cells, and mesenchymal cells classified depending on the tissue of origin. The results obtained have been grouped based on the cell lines′ tissue of origin, as the lung, skin, colon, pancreas, and neuro-ectoderm. The color table shows the magnitude of the mRNA expression levels assigned as reported in the legend and obtained upon the normalization of each sample with the *ACTB2* gene. The drawings accompanying the table illustrate the cell lines′ origin. Created in BioRender.com.

**Figure 5 ijms-26-03617-f005:**
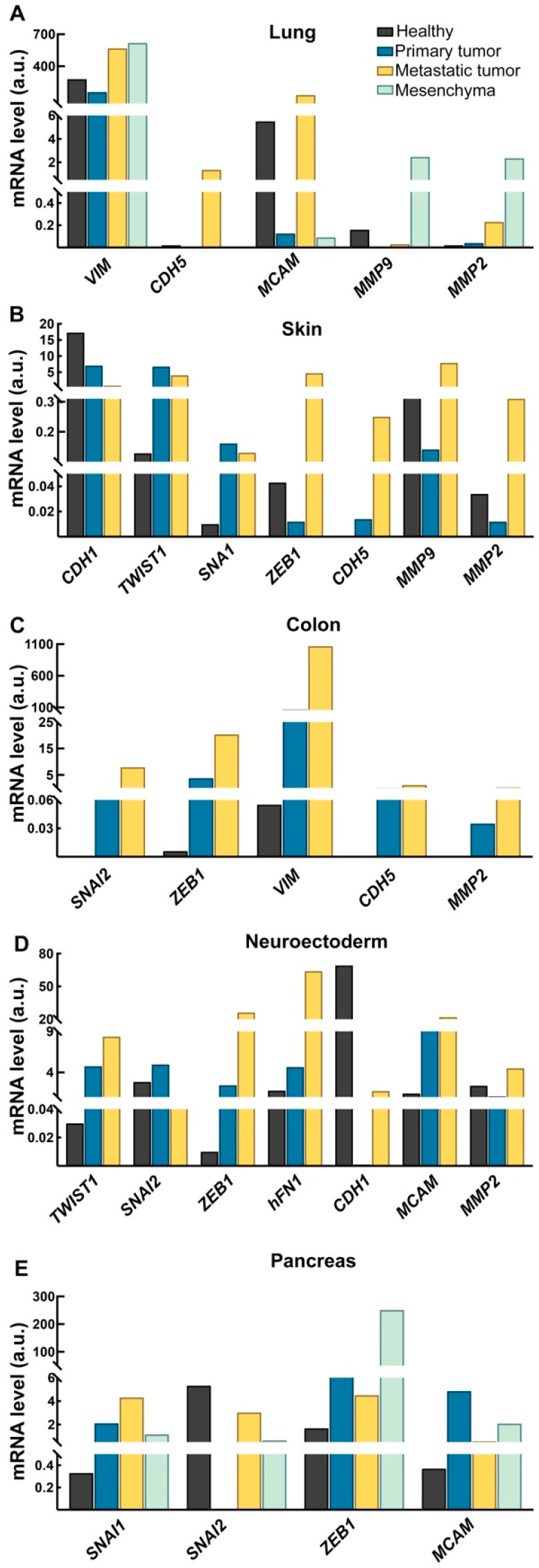
Bar graph representation of the tissue-specific levels of the EMT hallmarks across healthy epithelial cells, primary and metastatic cancer cell lines, and mesenchymal cells. The data obtained on the expression level of each gene have been grouped depending on the cells′ tissue of origin. (**A**): Lung, (**B**): Skin; (**C**): Colon; (**D**): Neuroectoderm; (**E**): Pancreas. The represented genes are the ones showing modulation during the transition from healthy to cancerous cells.

**Table 1 ijms-26-03617-t001:** *CDH5* genes and their ratio.

**Gene**	**Healthy vs. Primary Tumor**				
	**mRNA levels** (**a.u.**)	**ROC AUC**	**Sensitivity** (**%**)	**Specificity** (**%**)	***p* value**
** *CDH1* **	>3.085	0.96	100	90	0.002
** *CDH1/ZEB1* **	<4.552	0.90	87.5	100	0.019
** *CDH5/CDH1* **	>0.001807	0.87	87.5	85.7	0.018
**Gene**	**Healthy vs. Metastatic Tumor**				
	**mRNA levels** (**a.u.**)	**ROC AUC**	**Sensitivity** (**%**)	**Specificity** (**%**)	***p* value**
** *ZEB1* **	>0.2765	1	100	100	0.009
** *CDH5* **	>0.135	0.92	100	80	0.028
** *CDH5/CDH1* **	>0.06031	1	100	100	0.005
** *CDH1/ZEB1* **	>4.186	1	100	100	0.009
**Gene**	**Primary vs. Metastatic Tumor**				
	**mRNA levels** (**a.u.**)	**ROC AUC**	**Sensitivity** (**%**)	**Specificity** (**%**)	***p* value**
** *CDH5* **	>0.1865	0.93	100	77.8	0.009
** *CDH5/ZEB1* **	<0.05273	0.88	87.5	100	0.028

**Table 2 ijms-26-03617-t002:** Cell lines included in the present study. Healthy cell lines, primary and metastatic tumor-derived cells, and mesenchymal cell lines, grouped based on their tissue of origin.

Origin	Cell Line	Status
**HUMAN LUNG**	**Beas B2**	Healthy Bronchial Epithelium
	**A-549**	Primary Lung Carcinoma
	**H460**	Metastatic Lung Carcinoma
**HUMAN COLON**	**CCD 841 CoN**	Healthy Colon Epithelium
	**HCT 116**	Primary Colon Carcinoma
	**SW 480**	Primary Dukes C Colon–Rectal cancer
	**SW 460**	Metastatic Dukes C Colon–Rectal cancer
**HUMAN PANCREAS**	**HPaEpiC**	Healthy Pancreas Epithelium
	**Panc1**	Primary Epithelioid Pancreatic carcinoma
	**MiaPaCa-2**	Primary Pancreatic Carcinoma
	**CFPAC-1**	Metastatic Pancreatic Carcinoma
**HUMAN SKIN**	**HEK** (**n**)	Healthy Neonatal Epidermal Keratinocytes
	**A431**	Primary Epidermoid Carcinoma
	**NMSC**	Non-Melanoma Skin Cancer
**HUMAN** **NEURO-ECTODERMA**	**NHEM**	Normal Human Epidermal Melanocytes
	**WM 115**	Primary Melanoma
	**A375**	Achromic Primary Melanoma
	**WM 266.4**	Metastatic Melanoma
**HUMAN** **ENDOTHELIUM**	**HUVEC**	Human Umbilical Endothelial cells
**HUMAN** **MESENCHYMA**	**NHLF**	Stromal Lung Fibroblasts
	**StromPA**	Stromal Pancreatic cells
	**EDS**	Human Skin Fibroblasts
	**CCD-1123SK**	Human Skin Fibroblasts
	**V140-HD**	Healthy Bone Marrow Donor-Derived Mesenchymal Cells [83]
	**V126-HD**	Healthy Bone Marrow Donor-Derived Mesenchymal cells [83]

**Table 3 ijms-26-03617-t003:** RT-qPCR primers. Sequences were reported ^§^ [47] or designed * from the predicted sequence using the Gene Scan program http://genes.mit.edu/GENSCAN.html accessed on 1 January 2022.

Gene	Primer Sequences
***CD146/MCAM* *****	F: 5′-AGCTCCGCGTCTACAAAGC-3′
	R: 5′-CTACACAGGTAGCGACCTCC-3′
***CDH1* ^§^**	F: 5′AAAGGCCCATTTCCTAAAAACCT-3′
	R: 5′TGCGTTCTCTATCCAGAGGCT-3′
***CDH2* ^§^**	F: 5′CTCCTATGA GTGGAA CAG GAA CG-3′
	R: 5′-TTG GAT CAA TGT CAT AAT CAA GTG CTGTA-3′
** *CDH5 ** **	F. 5′-CACTGGAACCCCCACAGGAAAAGA-3′
	R. 5′-GGACAGCGTTCTCACACACTTTGG-3′
***HFN1* ^§^**	F: 5′-AGCCGAGGTTTTAACTGCGA-3′
	R: 5′-CCC ACT CGGTAAGTGTTCCC-3′
***VIM* ^§^**	R: 5′-GACGCCATCAACACCGAGTT-3′
	F: 5′-CTTTGTCGTTGGTTAGCTGGT-3′
***SNAI1* ^§^**	F: 5′-CCCAGTGCCTCGACCACTAT-3′
	R: 5′-GCTGGAAGGTAAACTCTGGATTAGA-3′
***SNAI2/SLUG* ^§^**	R: 5′-CCAAGCTTTCAGACCCCCAT-3′
	F: 5′-GAAAAAGGCTTCTCCCCCGT-3′
***TWIST1* ^§^**	R: 5′- GCTTGAGGGTCTGAATCTTGCT-3′
	F: 5′-GTCCGCAGTCTTACGAGGAG-3′
***ZEB1* ^§^**	R: 5′-CAGCTTGATACCTGTGAATGGG-3′
	F: 5′-TATCTGTGGTCGTGTGGGACT-3′
** *MMP2 ** **	F: 5′-CCTGCCCCTCCCTTCAACCA-3′
	R: 5′-GTTTCCGCTTCTGGCTGGGTC-3′
** *MMP9 ** **	F: 5′-CGGAGTGGCAGGGGGAAGATG
	R. 5′-CGGAGTGGCAGGGGGAAGATG
** *ACTB ** **	R: 5′-GAGACCTTCAACACCCCAGCC-3
	R. 5′-AATGTCACGCACGATTTCCC-3′

## Data Availability

The original contributions presented in this study are included in the article. Further inquiries can be directed to the corresponding authors.

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
