# Peer review of "The Clinical Relevance of Epithelial-to-Mesenchymal Transition Hallmarks: A Cut-Off-Based Approach in Healthy and Cancerous Cell Lines"

_ijms, 2025, doi:10.3390/ijms26083617_

Round 1

Reviewer 1 Report

Comments and Suggestions for Authors

The study explores the expression of EMT genes as cells transition across various cellular states towards metastatic seeding. The study identified CDH1, CDH5, and ZEB1 gene signature through qRT-PCR methods, capable of separating primary and metastatic cells. Expression was compared between mesenchymal-derived CTCs and a range of other primary and metastatic cell lines (25 in total). There are several concerns with this manuscript, primarily stemming from how gene expression was analysed and shown as a.u. - it is not clear how these values were generated and compared. Secondly, all the figures are derived from the same dataset, simply a selective mRNA expression screen by qRT-PCR, summarised in figure 1, just expressed or analysed in a different way. This limits the validity and robustness of the study, as does the small sample size and unknown repeatability of the assays. As such all results should be part of the same figure. Perhaps a redesign of the figures to break the comparisons down based on tissue-of-origin cell types might allow the findings to be conveyed in a more scientifically acceptable manner. Finally, as all the data is just evaluation of mRNA data, derived from a small number of sample types and with no information on reproducibility, the mechanistic insights offered are limited and the attempts to derive meaning from the expression comparison are lacking in robustness. As such, the manuscript is not acceptable in its current form.  

Further comments below: 

The use of the term primary tumour cells (i.e., Figure 1 (labelled as Prymary) and line 188) is misleading – they are well established, immortalised tumour cell lines, not primary or patient-derived. Clarifying that primary refers to tumour site of origin (as indicated in the Figure 1 schematic and legend), in contrast to cell lines considered to be derived from metastases, is needed in the main text body. Describing HUVECs as mesenchymal is also misleading.  

It is not clear what the a.u. values shown in figure 1 to describe expression levels refers to or how it was derived/determined. Why were the values, apparently analysed using the ΔΔCt method, not expressed as log2 or log10 fold changes against a standard control line? How the a.u. were determined/derived is not clear or explained in the methods. How the cut-offs were determined as shown in Table 1 was not explained.  

It is also curious and unexpected that the human skin fibroblast analysed in Figure 1 express more VE-cadherin than HUVECs – this required comment, as both fibroblasts examined appear to express CDH5 mRNA. Phrase on line 209 (fundamental to sustain cancer cells proliferation stimulating angiogenesis) also incorrect regarding VE-cadherin – which is only required for endothelial cell junctional stabilisation and angiogenesis, not cancer cell proliferation (indeed, it is largely not expressed in cancer cells). The description of CDH5 expression accompanying Figure 1 is misleading when the results in Figure 2 show only a slight change in expression (shown as a.u) between the 4 cellular types compared. The mesenchymal cells expression of CDH5 appears highly skewed – perhaps by the fibroblasts and HUVECs – and should not be described as a marker of aggressiveness, as suggested in line 207. It should be the comparisons between cells derived from the site of origin versus metastases that should be the focus.  

It is largely well written, and narrative logically developed. Penultimate introductory paragraph beginning at 137-144 needs editing to remove repetitive terminology around gene annotations. Line 143-144 statement (overall supporting EMT activation as the cause for metastasis and resistance to chemotherapy) is unsupported by the current study and should be modified). Some additional minor corrections are noted below. 

Sufficient background to the main EMT processes is provided within the introduction, supported by several clinical samples. Specifically describing EMT-associated genes regulated by SNAI1, ZEB1, and TWIST1 would enhance understanding of the molecular drivers and link to paragraph beginning at line 125.  

How figure 2 was generated is unclear. What values were used to make the violin plots needs clarification. Justification of why a.u values were used for expression comparison needed. How the normalisation was carried out to get the range of ''a.u'' from 0.1 to 3000+ is unclear. 

Authors were careful to point out the limitations of using ROC on small sample numbers and that care should be taken with interpretation.  

As the same data is used in Figure 1 and 4, the utility is limited, with figure 4 adding little to the arguments.  

Similarly Figure 5 appears to be a re-interpretation of the same data used in the preceding figures and lacks robustness.  

Many of the attempted correlative analyses are speculative given they are based on 1-4 cell lines.  

No information provided on how many different samples - ''n'' - were used to generate the cDNA and expression data. 

It is not clear why the cut-off values were chosen/derived and how they are discriminatory, given the small sample numbers – i.e., line 334-336 in the discussion - ''The cut off values for CDH1 able to differentiate non-cancer from primary tumor cells is 0.385 a.u, whilst the CDH1/ZEB1 and the CDH5/CDH1 ratios discriminating healthy from primary tumor cells were <4.552 and >0.001807 a.u., respectively.'' The data used and how it is presented makes it difficult to validate/evaluate.  

Minor changes: 

VE should be vascular endothelial (103).  

Clarifying that VE-cadherin is expressed within the melanoma tumour microenvironment/stroma is needed (105).  

VEGFR2 should be described as vascular endothelial growth factor receptor 2 (KDR), not VE growth factor receptor 2 (107).  

Statement on line 109 (following binding of the EMT-TF TWIST1) requires clarification, as TWIST does not bind to VE-cadherin protein (but rather the promoter) as suggested by the phrase.  

Combine [40][41] on 113.  

Line 151 should be in past tense. 

Whit should be with – line 227. 

Numerous grammatical and spelling errors in the discussion.  

Author Response

       First, I would like to thank the reviewers for their thorough analysis of our manuscript and for their valuable suggestions to improve the quality of our work.

We will now provide a point-by-point response to Reviewer 1's comments.

All the changes introduced in the main text have been highlighted in red.

Answer to Reviewer 1

  1. The study explores the expression of EMT genes as cells transition across various cellular states towards metastatic seeding. The study identified CDH1, CDH5, and ZEB1 gene signature through qRT-PCR methods, capable of separating primary and metastatic cells. Expression was compared between mesenchymal-derived CTCs and a range of other primary and metastatic cell lines (25 in total)”.

We would like to clarify that the mesenchymal cell lines included in the present study are not derived for purified CTCs. Most of the cell lines have been purchased from ATCC (The American Type Culture Collection) as reported in the Material and Methods section (pag. 14, lines 425-427), apart from two mesenchymal cell lines originating from two bone marrow healthy donors (pag 14, lines 436-441). The purchased cells were “cell-line”-authenticated and certified for bacteria and virus absence. All the cell lines were analyzed by PCR to identify a possible Mycoplasma contamination; at maximum passage 3 cells were detached and the RNA was analyzed (lines 454-462).

  1. “There are several concerns with this manuscript, primarily stemming from how gene expression was analysed and shown as a.u. - it is not clear how these values were generated and compared”.

                In the Materials and Methods section, we stated that “Gene expression normalized to β-actin was calculated using the 2−ΔCt method” (page 16, line 496). Specifically, three independent cultures of each cell line were analyzed to assess the expression levels of the genes included in the panel. Each RT-qPCR run was performed in duplicate for each gene. The ACTB gene, encoding β-actin, was used as an internal reference gene. The mRNA levels of each target gene were normalized to their respective ACTB levels. For this reason, the obtained values are presented as arbitrary units (a.u.), as the Ct values of the genes of interest were normalized to the Ct values of the ACTB gene from the same sample.

  1. “Secondly, all the figures are derived from the same dataset, simply a selective mRNA expression screen by qRT-PCR, summarised in figure 1, just expressed or analysed in a different way. … As such all results should be part of the same figure. Perhaps a redesign of the figures to break the comparisons down based on tissue-of-origin cell types might allow the findings to be conveyed in a more scientifically acceptable manner”.

                We appreciate Reviewer 1 suggestion; however, we believe that our data representation can best showcase the two sides of the same coin. Indeed, we would like to highlight the complexity of the EMT process. Just as Reviewer 1 correctly observed, analyzing the same variables from two different perspectives leads to extremely different conclusions. In Figure 1, if we examine how epithelial/mesenchymal marker levels change during the transformation from healthy to neoplastic cells, we conclude that, regardless of the tissue of origin, there is a significant variation in the genes CDH1, TWIST1, SNAI1, hFN1, and MMP2. This could suggest the validation of these genes as markers of tumor progression. On the contrary, when analyzing their variation based on the tissue of origin, the information we obtain is completely different. It appears that some of these genes lose significance, while others, such as VIM and ZEB1 in the colon, gain relevance. We added these considerations in the discussion section to clarify the importance of both these analyses (pag. 13, lane 378-385).

Regarding the repeatability of the assay, we culture cells independently for three times. We added this information in the Material and Methods section, under the sub-section 4.1. Cell lines (lines 454-462).  

  1. Finally, as all the data is just evaluation of mRNA data, derived from a small number of sample types and with no information on reproducibility, the mechanistic insights offered are limited and the attempts to derive meaning from the expression comparison are lacking in robustness. As such, the manuscript is not acceptable in its current form.  

We thank Reviewer 1 for her/his suggestion to investigate this phenomenon from a mechanistic point of view. However, although the mechanism by which EMT promote and improve tumor progression and dissemination is well established (https://doi.org/10.1186/s13045-022-01347-8) as well as its clinical relevance (https://doi.org/10.1016/j.canlet.2015.06.007), the investigation of the mechanism(s) underlying cancer dissemination via EMT is not the purpose of the present study. As stated in the manuscript title, “a cut-off based approach,” our aim is to explore, for the first time to the best of our knowledge, potential mRNA cut-off values for the investigated EMT hallmarks to evaluate cancer aggressiveness and metastatic potential. In the future, we intend to extend our study to RNA obtained from patient-derived tumor tissues as well as circulating tumor cells (CTCs). Concerning reproducibility, All the experiments (cell cultures) were performed in triplicate for each tissue it was analyzed, for the healthy condition, primary- and metastatic tumor derived cells and the mesenchymal cell lines. We added this information in Material and Methods section (lines 454-462).

Further comments below: 

  1. The use of the term primary tumour cells (i.e., Figure 1 (labelled as Prymary) and line 188) is misleading – they are well established, immortalised tumour cell lines, not primary or patient-derived. Clarifying that primary refers to tumour site of origin (as indicated in the Figure 1 schematic and legend), in contrast to cell lines considered to be derived from metastases, is needed in the main text body”.

                We agree with the reviewer and, accordingly, we modified the main text.

  1. “Describing HUVECs as mesenchymal is also misleading”.

We agree with the reviewer and apologize for the misunderstanding. Indeed, HUVEC has been classified as human endothelial cells (Materials and Methods, pag. 15, Table II). However, their representation as last cell line in the group of the mesenchymal cells is misleading. Accordingly, we have modified Figure 1.

  1. “It is not clear what the a.u. values shown in figure 1 to describe expression levels refers to or how it was derived/determined. Why were the values, apparently analysed using the ΔΔCt method, not expressed as log2 or log10 fold changes against a standard control line? How the a.u. were determined/derived is not clear or explained in the methods”.

As stated in our response to point 2, the expression levels reported in arbitrary units refer to the results obtained after normalization with the ACTB2 gene assayed in each sample, following the 2-ΔCt method, as reported in the Material and Methods section, pag. 15, lines 501-504). Therefore, we did not use the ΔΔCt method or report the results as fold changes. The rationale behind this choice is that, given the main objective of our study—namely, the possible identification of cut-off values for the mRNA expression of EMT hallmarks—it would not be feasible to express their levels as "fold change" relative to a healthy control. In clinical practice, none of the analytes assessed for diagnostic or prognostic purposes are expressed in fold change; instead, they are reported in units such as ng/dL, U/mL, or mg/dL. Furthermore, our future goal is to apply the putative cut-off values to circulating tumor cells (CTCs), for which a healthy control will never be available, as healthy individuals do not have detectable CTCs.

  1. “How the cut-offs were determined as shown in Table 1 was not explained”.

We established the cut-off values presented in Figure 1 based on the mRNA expression levels of the analytes investigated in healthy cell lines, which served as our reference point. Since the healthy cell lines included were primarily epithelial, the cut-off values were determined by considering these cells as expressing high levels of epithelial markers (e.g., CDH1) and low levels of mesenchymal markers (e.g., hFN1, CDH2, and MMPs). Then, we used mesenchymal cells as our positive control for the mRNA expression of mesenchymal genes, including hFN1, TWIST1, SNAI1, MCAM, and MMP2. To better clarify this point we added this part in the main text (pag. 4, lines 179-183).

  1. “It is also curious and unexpected that the human skin fibroblast analysed in Figure 1 express more VE-cadherin than HUVECs – this required comment, as both fibroblasts examined appear to express CDH5 mRNA”.

We totally, agree with the Reviewer that pointed out a transcription error of the results obtained

from the skin-derived fibroblasts regarding the mRNA expression of the CDH5 gene. Accordingly, both Figure 1 and Figure 2 have been modified.

  1. “Phrase on line 209 (fundamental to sustain cancer cells proliferation stimulating angiogenesis) also incorrect regarding VE-cadherin – which is only required for endothelial cell junctional stabilisation and angiogenesis, not cancer cell proliferation (indeed, it is largely not expressed in cancer cells)”.

We thank the reviewer for his/her suggestion. Hower, we would like to point out that the role of Ve-cadherin is further beyond that of “endothelial cell junction stabilization”. Metastasis initiation is driven by cellular plasticity, including processes such as EMT and MET. Cell-cell adhesion molecules, such as cadherins, play a critical role in regulating cancer invasion and metastasis, influencing not only adhesion but also cell migration. E-cadherin and N-cadherin are considered the primary cadherins involved in EMT, particularly in the “cadherin switch”. However, other cadherins, such as Cadherin-5, 6, and 17, are also known to contribute to cancer progression and metastasis, showing overexpression across a diverse range of solid tumors. A bulk of literature has demonstrated the implication and the requirement of VE-cadherin in cancer cell proliferation (10.18632/oncotarget.13832; https://doi.org/10.1186/bcr3367, https://doi.org/10.3390/ijms222413358) vasculogenic mimicry (https://doi.org/10.1073/pnas.131209798, https://doi.org/10.1186/s12943-017-0631-x,) and its expression in cancer cells [breast cancer (10.1007/s00418-017-1619-8); melanoma (https://doi.org/10.1073/pnas.131209798); esophageal carcinoma (10.3748/wjg.v20.i47.17894); salivary gland (10.4103/jispcd.JISPCD_323_17)]. However, we agree with Reviewer I that the term “angiogenesis” was inappropriately used, and we replace it with “vasculogenic mimicry” (pag. 6, lane 220).

  1. The description of CDH5 expression accompanying Figure 1 is misleading when the results in Figure 2 show only a slight change in expression (shown as a.u) between the 4 cellular types compared. The mesenchymal cells expression of CDH5 appears highly skewed – perhaps by the fibroblasts and HUVECs – and should not be described as a marker of aggressiveness, as suggested in line 207”.

We apologize for any confusion regarding the representation of CDH5 expression in Figures 1 and 2. However, as stated in our response to point 6, HUVEC was not included in the mesenchymal cell group. As explained in our response to point 3, the two figures present different perspectives. Figure 1 groups cell lines based on their "aggressiveness" representing each single value obtained, ranging from healthy to mesenchymal cells. In contrast, Figure 2 represents the mean of the mRNA levels obtained in each of the four group (healthy, primary-tumor derived cells, metastatic-tumor derived cells and mesenchymal cells). however, following the correction of the mistake in reporting the CDH5 expression in skin-derived fibroblasts, the graph slightly changed.

  1. “It should be the comparisons between cells derived from the site of origin versus metastases that should be the focus”.

We would like to thank the reviewer for his/her suggestion; however, we do not fully understand his/her point of view. Thus, our goal is the identification and the clinical interpretation of minimal residual disease through the purification and molecular characterization of factors related to tumor aggressiveness, such as EMT, cell-cell-adhesion molecules and invasiveness-associated factors. Before facing the complexity of CTCs we decide to establish putative cut-off of the investigated genes starting from healthy-tissues derived cells to fully mesenchymal cells to evaluate possible markers describing cancer aggressiveness.

Of note, in the present manuscript we compared not only the results obtained from healthy cells to those from primary-cancer derived ones, but as also suggested by the reviewers, we included the comparison between healthy and metastases-derived cancer cells, as well as primary tumor-derived cells versus metastatic-tumor derived cells. Of note, the release of CTCs occurs after transformed mesenchymal-like cells detach from the primary tumor mass, following the fulfillment of EMT. Therefore, it could be just as valuable to analyze the differential expression of EMT markers not only between healthy and metastatic tumor-derived cancer cells but also between heathy and primary and also between healthy and metastatic tumor-derived cells

  1. “Penultimate introductory paragraph beginning at 137-144 needs editing to remove repetitive terminology around gene annotations”.

Duplications removed.

  1. “Line 143-144 statement (overall supporting EMT activation as the cause for metastasis and resistance to chemotherapy) is unsupported by the current study and should be modified)”.

Regarding this point, we would like to emphasize that our manuscript is the latest in a body of evidence supporting the crucial role of EMT in tumor progression and the promotion of the metastatic cascade (10.1016/j.tranon.2020.100773). The added value of our work lies in conducting a preliminary study aimed at identifying potential cut-off values that could enable the introduction of EMT markers into clinical practice.

  1. “Specifically describing EMT-associated genes regulated by SNAI1, ZEB1, and TWIST1 would enhance understanding of the molecular drivers and link to paragraph beginning at line 125”.

The text has been modified accordingly to reviewer 1’s requests (pag. 2, lane 65-74).

  1. “How figure 2 was generated is unclear. What values were used to make the violin plots needs clarification. Justification of why a.u values were used for expression comparison needed. How the normalisation was carried out to get the range of ''a.u'' from 0.1 to 3000+ is unclear”.

Please, see answer to point 7 and 8.

  1. “As the same data is used in Figure 1 and 4, the utility is limited, with figure 4 adding little to the arguments. Similarly Figure 5 appears to be a re-interpretation of the same data used in the preceding figures and lacks robustness.

Please, see response to point 3.

  1. “No information provided on how many different samples - ''n'' - were used to generate the cDNA and expression data”.

We agree with the reviewer and apologize for not including this information in the Materials and Methods section. We have now completed this part by adding the missing details (page 15, lines 488–490 and 515–516).

  1. “It is not clear why the cut-off values were chosen/derived and how they are discriminatory, given the small sample numbers – i.e., line 334-336 in the discussion - ''The cut off values for CDH1 able to differentiate non-cancer from primary tumor cells is 0.385 a.u, whilst the CDH1/ZEB1 and the CDH5/CDH1 ratios discriminating healthy from primary tumor cells were <4.552 and >0.001807 a.u., respectively.'' The data used and how it is presented makes it difficult to validate/evaluate”.

We agree with Reviewer 1 that, at present, applying these cut-off values without proper validation would be challenging, at least in patient-derived tissues, where the presence of a healthy control for comparison would be possible. Certainly, the cut-off value obtained for the expression of a single gene will differ from that derived from the ratio of two genes.

  1. “VE should be vascular endothelial (103)”.

Amendend (lane 110).

  1. “Clarifying that VE-cadherin is expressed within the melanoma tumour microenvironment/stroma is needed (105)”.

Clarified at pag. 3, lane 113-116.

  1. “VEGFR2 should be described as vascular endothelial growth factor receptor 2 (KDR), not VE growth factor receptor 2 (107)”.

Amended.

  1. “Statement on line 109 (following binding of the EMT-TF TWIST1) requires clarification, as TWIST does not bind to VE-cadherin protein (but rather the promoter) as suggested by the phrase”.

Amended.

  1. “Combine [40][41] on 113”.

Amended

  1. “Line 151 should be in past tense”.

If we understood correctly, Reviewer 1 is referring to the paragraph starting with “In light of this evidence, we propose to …”. We believe that the study's purpose should be presented using the present tense.

  1. “Whit should be with – line 227”.

Amended

  1. “Numerous grammatical and spelling errors in the discussion”.

Amended

Reviewer 2 Report

Comments and Suggestions for Authors

The manuscript investigates the expression of epithelial-to-mesenchymal transition (EMT)-related genes across healthy, primary, and metastatic cancer cells from various tissues. The study aims to establish cut-off values for specific EMT markers to assess cancer aggressiveness and potential therapeutic response. The authors analyze the modulation of EMT hallmarks in different cell lines and propose tissue-specific EMT signatures. While the study offers interesting insights, several issues need attention before recommending it for publication.

Major points need to be addressed:

  1. Sample Size and Cell Line Maintenance:

    The study relies on a relatively small number of cell lines (5–8 per condition), which may limit the generalizability of the findings. Contamination of cell lines is a common issue in scientific research. Could the authors provide information on the maintenance of these cell lines? It is essential to report the number of passages at the time experiments were conducted and include a recent STR profile to confirm no contamination.

  2. The methods/results section should clarify how Figures 1, 2, 4, and 5 were generated. Are they derived from the same dataset but presented differently?

  3. In Figure 1, variability for some genes is high, and the current scoring system lacks detailed information. It is recommended to present real fold changes (e.g., means ± SD) while retaining the color scale (yellow-pink) to visualize trends.
  4. In Figure 2, the violin plot with a break in the axis appears misleading. Given that each group has no more than eight data points, a violin plot is unnecessary. A log-scale plot would be more appropriate.

  5. In Figure 3, the ROC analysis using only a limited number of cell lines is unreliable. Future validation in patient-derived tissues is strongly recommended.

  6. The concerns regarding Figure 1 also apply to Figure 4.

  7. In Figure 5, there is no statistical analysis, yet the authors claim up/downregulation of related genes. This weakens the reliability of the results.

Minor points need to be addressed:

Inconsistent font and size appear throughout the manuscript (e.g., Sections 4.1 and 4.2). Standardization is necessary.

Author Response

Answer to Reviewer 2

First, I would like to thank the reviewers for their thorough analysis of our manuscript and for their valuable suggestions to improve the quality of our work. We will now provide a point-by-point response to Reviewer 1's comments. All the changes introduced in the main text have been highlighted in red.

  1. “Contamination of cell lines is a common issue in scientific research. Could the authors provide information on the maintenance of these cell lines? It is essential to report the number of passages at the time experiments were conducted and include a recent STR profile to confirm no contamination”.

We thank Reviewer 2 for his/her suggestion. We added the requested informations in the Material and Methods section (pag.

Cells were placed in cultures and subjected to a maximum of three passages and then detached by trypsinization, centrifuged, washed twice with phosphate buffered saline (PBS) and stored at –70°C, until RNA extraction. The culture media as reported in the material and methods section all contained antibiotics. The purchased cells were cell-line Authenticated (CLA) and certified for bacteria and virus absence. They were all been subjected to mycoplasma test (added in the material and methods text, sub-section 4.1. Cell lines (lines 454-462).  

  1. “The methods/results section should clarify how Figures 1, 2, 4, and 5 were generated. Are they derived from the same dataset but presented differently?”

The data are derived from the same dataset but highlight the complexity of the EMT process, which can vary significantly depending on the tissue-specific EMT gene signature. We stated that:

  • To better define and visualize the possible changes occurring in the EMT hallmarks during the different phases of cancer, we also represented the results as bar graphs of the averaged level of each marker obtained in healthy epithelial cells, primary and metastatic tumor and in the mesenchymal cells (Figure 2) (pag. 4, lane 202-204).
  • To investigate the possible tissue-specificity of the EMT markers under investigation, the results obtained from the healthy, primary and metastatic cancer cells models were divided based on the tissues of origin (Figure 4) (pag. 8, lane 261-263). As it is also possible to observe in Fig 2 the cell lines analyzed are the same reported in Figure 1.
  • However, to better visualize these distributions, we presented the collected data also as bar graphs (Figure 5) (pag. 8, lane 270-271).
  1. “In Figure 1, variability for some genes is high, and the current scoring system lacks detailed information. It is recommended to present real fold changes (e.g., means ± SD) while retaining the color scale (yellow-pink) to visualize trends”.

We added the information on the elaborated scoring system at pag. 4, lane 186-190). The values are presented as arbitrary unit (a.u.) and not in fold change, following the 2-DCt method and not the 2-DDCt method. In particular, the expression levels reported in arbitrary units refer to the results obtained after normalization with the ACTB2 gene assayed in each sample, as reported in the Material and Methods section, pag. 15, lines 501-504). Therefore, we did not use the ΔΔCt method or report the results as fold changes. The rationale behind this choice is that, given the main objective of our study—namely, the possible identification of cut-off values for the mRNA expression of EMT hallmarks—it would not be feasible to express their levels as "fold change" relative to a healthy control. In clinical practice, none of the analytes assessed for diagnostic or prognostic purposes are expressed in fold change; instead, they are reported in units such as ng/dL, U/mL, or mg/dL. Furthermore, our future goal is to apply the putative cut-off values to circulating tumor cells (CTCs), for which a healthy control will never be available, as healthy individuals do not have detectable CTCs.

  1. “In Figure 2, the violin plot with a break in the axis appears misleading. Given that each group has no more than eight data points, a violin plot is unnecessary. A log-scale plot would be more appropriate”.

We accordingly modified Figure 2. However, it has not been possible to express the value in a log scale since the “undected” genes expressions, corresponding to 0 arbitrary unit value, will not be considered. Additionally, we found an error in the reported value for CDH5 gene level obtained from skin-derived fibroblasts.

  1. “In Figure 3, the ROC analysis using only a limited number of cell lines is unreliable. Future validation in patient-derived tissues is strongly recommended”.

We totally agree with Reviewer 2. For this reason, we stated that: “Although it is known that an accurate ROC analysis requires a much larger number of samples [49], we believe that this can still be a useful preliminary approach to identify possible diagnostic cut-offs to further deepen the study”.

  1. “The concerns regarding Figure 1 also apply to Figure 4”.

See response to point 3

  1. In Figure 5, there is no statistical analysis, yet the authors claim up/downregulation of related genes. This weakens the reliability of the results.

The lack of statistical analyses is due to the number of cell lines analyzed in each group (healthy, primary tumor-derived, metastatic tumor-derived, and mesenchymal cells). Not all groups included at least three different cell lines, which is the minimum required for a reliable statistical analysis. We added this explanation at pag. 8, lane 271-275. Thus, in accordance with the Reviewer’s suggestion, we have revised the Results section to discuss the data in terms of a “trend” rather than defining them as up- or down-regulated.

  1. Inconsistent font and size appear throughout the manuscript (e.g., Sections 4.1 and 4.2). Standardization is necessary.

Amended.

Round 2

Reviewer 2 Report

Comments and Suggestions for Authors

The authors has addressed most of my concerns and the revised manuscript is publishable.

Only few minor typos need to be addressed:

Line 451,CO2 should be CO2

Authors contribution, please modify the fonts to be consensus.